# Exogenous Rubella Virus Capsid Proteins Enhance Virus Genome Replication

**DOI:** 10.3390/pathogens11060683

**Published:** 2022-06-14

**Authors:** Min-Hsin Chen, Cara C. Burns, Emily Abernathy, Adaeze A. Ogee-Nwankwo, Joseph P. Icenogle

**Affiliations:** 1Division of Viral Diseases, Centers for Disease Control and Prevention, Atlanta, GA 30333, USA; zqd1@cdc.gov (C.C.B.); bioesa@yahoo.com (E.A.); jnmadice2@gmail.com (J.P.I.); 2Center for Global Health, Centers for Disease Control and Prevention, Atlanta, GA 30333, USA; fyz7@cdc.gov

**Keywords:** rubella virus, capsid, exogenous, genome replication

## Abstract

Enhanced replication of rubella virus (RuV) and replicons by de novo synthesized viral structural proteins has been previously described. Such enhancement can occur by viral capsid proteins (CP) alone in trans. It is not clear whether the CP in the virus particles, i.e., the exogenous CP, modulate viral genome replication. In this study, we found that exogenous RuV CP also enhanced viral genome replication, either when used to package replicons or when mixed with RNA during transfection. We demonstrated that CP does not affect the translation efficiency from genomic (gRNA) or subgenomic RNA (sgRNA), the intracellular distribution of the non-structural proteins (NSP), or sgRNA synthesis. Significantly active RNA replication was observed in transfections supplemented with recombinant CP (rCP), which was supported by accumulated genomic negative-strand RNA. rCP was found to restore replication of a few mutants in NSP but failed to fully restore replicons known to have defects in the positive-strand RNA synthesis. By monitoring the amount of RuV RNA following transfection, we found that all RuV replicon RNAs were well-retained in the presence of rCP within 24 h of post-transfection, compared to non-RuV RNA. These results suggest that the exogenous RuV CP increases efficiency of early viral genome replication by modulating the stage(s) prior to and/or at the initiation of negative-strand RNA synthesis, possibly through a general mechanism such as protecting viral RNA.

## 1. Introduction

Rubella virus (RuV) is a member of the Rubivirus genus within the Matonavirus family [1]. RuV is an enveloped virus with a single-stranded RNA genome of positive polarity. The genome is usually 9762 nucleotides (nt) in length [2], 5′ capped and 3′ poly-adenylated and encodes two open reading frames (ORF): the 5′ ORF encodes the viral nonstructural proteins (NSP) and the 3′ ORF encodes the viral structural proteins (SP), including the capsid proteins (CP), and two envelope glycoproteins, E2 and E1. In the infection cycle, after uncoating, the replication of RuV starts with the translation of the viral NSP from genomic RNA (gRNA). The NSP polyprotein, which has an approximate total molecular weight (MW) of 220 kDa, is processed by a virally encoded protease into two proteins of MW 150- and 90-kDa. The p150 protein contains a domain homologous to methyltransferase and possesses protease activity [3,4]. The p90 exhibits nucleoside triphosphatase activity [5] and bears an RNA-dependent RNA polymerase (RdRp) domain [6]. Replication of the viral genome by the NSP begins with synthesis of the complementary strand of gRNA [(−) RNA]. This genomic complement then serves as a template for the synthesis of positive-strand genomic [(+) gRNA] and subgenomic RNAs (sgRNA). The sgRNA serves as mRNA for the synthesis of the viral SP. When the SP are used in an expression system, they can be assembled and released as virus-like particles in the absence of viral gRNA [7]. Only the (+) gRNA is packaged into virions [8].

Studies of RuV RNA replication were expedited after development of infectious cDNA clones and replicons [9,10,11]. Expression of reporter genes from RuV replicons with deletion of the coding regions for CP, E2, and part of E1 in cells transfected with replicon RNA has been described [10,11]. The mutant RUBrep/GFP_ΔNotI, which contains an in-frame deletion (nt 1693–2200; aa 548–717) near the 5′ end of the NSP coding region, was of high interest because GFP (green fluorescent proteins) expression from this mutant was detected only in the presence of infectious virus, while the same deletion, when introduced into the infectious viral RNA, still produced infectious virus [10]. It was found that RuV CP alone could enhance the GFP expression of this mutant [12,13]. CP was proposed to specifically trans-complement a region between aa 497–803 in the NSP, named Q domain, which overlapped with the NotI fragment [14]. RuV CP also modulated replication of virus and replicon genomes bearing mutations in the cis-acting elements [12,15]. The modulation depended on the amounts of input RNA and RuV CP, i.e., the enhancement was more significant with a smaller amount of transfected RNA. These studies suggest that RuV CP have different roles during different stages of the virus replication cycle: the CP not only forms nucleocapsids during viral assembly (late stage) [16] but also plays a role in viral genome replication enhancement (early stage) [12].

The dissociation of the RuV genome from the nucleocapsids after entry is not fully understood. It was suggested that the phosphorylation of RuV CP in the virions is critical in facilitating the release of viral genome from the nucleocapsids during disassembly, probably because of the less stable interaction between the CP and RuV RNA [17,18]. While little is known about RuV disassembly, the uncoating of alphaviruses has been well characterized and occurs through a ribosome-mediated pathway [19,20]. Once the alphavirus nucleocapsid is dissociated from the viral RNA, it is translocated to the cytoplasmic side of the endosomal membrane and subjected to cellular proteases. The disassembly of the RuV nucleocapsids may be substantially different from the alphaviruses because the RuV CP remain membrane-associated by their carboxyl-termini and the structure of RuV virions is different from that of classic icosahedrons [21]. For instance, the RuV CP may not be completely stripped from the viral RNA so that the remaining RuV CP is involved in RuV replication. The discovery of two new Matonaviruses [22], by offering more closely related viruses, may provide better models than alphaviruses.

The mechanism(s) by which RuV CP modulate genome replication is not fully understood. RuV CP have been found to colocalize with RuV NSP [23,24] or viral RNA [25], suggesting that intracellular CP are in close proximity to the viral replication complex. Nevertheless, none of these studies examined whether the exogenous RuV CP, i.e., CP from the input virus particles, could also participate in genome replication enhancement. In this study, we specifically explored the role of RuV CP in virions on genome replication enhancement during the early stage(s) of the virus infection cycle.

## 2. Results

### 2.1. Exogenous Virion Proteins Enhance RuV Genome Replication

Based on the previous finding that GFP expression was only detected in RuV-infected cells transfected with RUBrep/GFP_ΔNotI [10], a replicon-based diagnostic method consisting of transfecting RUBrep/GFP_ΔNotI RNA into cells inoculated with clinical specimens was proposed [26]. We attempted to simplify this detection method by delivering the same replicon using an RuV pseudovirus. Detection of GFP was expected only in cells infected with both the pseudovirus and wild-type virus and not in cells with the pseudovirus alone. However, a high level of GFP expression in cells infected with the replicon-containing pseudovirus was consistently observed in the absence of infectious virus (P1) (Figure 1A). This GFP expression was not observed when the supernatant from these cells was used to infect new Vero cells (P2) (data not shown). The absence of GFP expression in P2-infected cells indicated the absence of infectious virus in the pseudovirus stock—if GFP expression in the initial pseudovirus-infected cells was due to contaminating RuV in the pseudovirus preparation (e.g., recombinant virus), one would expect that the GFP expression from the replicon would have been detected when the supernatant was used to infect new Vero cells [27].

We hypothesized that the RuV SP in the pseudovirus particles were promoting the replication of RUBrep/GFP_ΔNotI after entering cells. To test this, RuV antigens, including UV-inactivated RuV virus (Meridian Life, Memphis, TN, USA) and virus-like particles (VLP; Advanced ImmunoChemical, Long Beach, CA, USA), were added to the Lipofectamine-RUBrep/GFP_ΔNotI RNA mixture during transfection. By the end of transfection, the monolayers were subjected to three PBS washes to remove excess external materials. As shown in Figure 1B, the addition of either UV-inactivated antigen or VLP resulted in GFP expression in cells transfected with RUBrep/GFP_ΔNotI. Expression of GFP decreased as the amount of VLP added decreased (data not shown). Other sources of RuV antigens, such as VLP from SP-expressing cells, also resulted in GFP expression in RUBrep/GFP_ΔNotI transfected cells (data not shown). No GFP expression was detected when UV-inactivated Sindbis virus (SINV) antigen was added to RUBrep/GFP_ΔNotI replicon (data not shown). No de novo synthesized SP or infectious virus was detected in pseudovirus-infected cells or cells transfected with replicons in the presence of RuV antigens by immunocolorimetric assay (ICA) or by RT-PCR, again ruling out the possibility of infectious virus being present from any source (data not shown). These results with exogenous RuV antigens suggest that the GFP expression using pseudovirus containing RUBrep/GFP_ΔNotI was the result of SP in the pseudovirus.

RuV RNA can replicate without supplemented SP [9]. To investigate whether the exogenous RuV virion proteins enhanced the replication of infectious RuV RNA synthesized in vitro, recovery of virus from the transfection with serially diluted infectious RuV RNA with and without RuV antigens (UV-inactivated RuV or RuV VLP) was compared. Infectious virus was recovered in cells transfected with 2 μg of viral RNA without supplemented RuV antigens (Figure 1C, bottom row). In the transfection with 0.2 μg infectious RNA, infectious virus was only recovered (>10*^3^* pfu/mL) if the RNA was mixed with VLP (Figure 1C, middle row). Although exogenous RuV proteins were not required to initiate infection, an approximately 3-fold increase of RuV titer was observed when RuV antigen was mixed with 1–2 μg of RuV RNA (3.0 ± 1.6) (Table 1). No sign of infectious virus was recovered in the transfection without viral RNA, even if VLP was added (Figure 1C, top row).

To test whether the exogenous RuV antigens specifically enhanced RuV replication, the production of other viruses, poliovirus (PV) and SINV, from Vero cells transfected with viral RNA with and without RuV antigens was compared. The viral RNAs, PV RNA transcribed in vitro from infectious cDNA clones [28] or total RNA extracted from SINV-infected cells, were transfected onto Vero cells and the amount of recovered virus in the presence of RuV antigens versus that in the absence of RuV antigens was determined by plaque assay at 24 to 72 h post-transfection. Unlike RuV, the addition of RuV antigens resulted in less virus production from PV and SINV transfected cells (Table 1), indicating that RuV antigens impacted viral replication at an intracellular level. In addition to viral RNA, the expression of Lac*Z* from a control DNA, pCH110 (GE Healthcare Life Sciences, Pittsburgh, PA, USA), with and without RuV antigen, was also compared. When RuV antigen was added to a transfection with the pCH110, no significant increase was observed in the intracellular β-galactosidase activity (relative activity of 0.91 ± 0.28) (Table 1). Taken together, exogenous RuV antigens specifically enhanced RuV genome replication.

### 2.2. The RuV CP Is the Major Component Enhancing RuV Replication

Previously, we have shown that de novo synthesized CP enhanced RuV genome replication [12]. To confirm that RuV CP can independently enhance RuV replication as shown in Figure 1, recombinant RuV proteins were mixed with RUBrep/GFP_ΔNotI during transfection. As shown in Figure 2, GFP expression was only detected when recombinant RuV CP (rCP) or whole antigens were added during transfection. No GFP expression was found in RUBrep/GFP_ΔNotI-transfected cells with recombinant E2 (rE2), E1 (rE1), or BSA (data not shown). The number of GFP-expressing cells also correlated with the amount of rCP used (data not shown). In the experiment shown in Figure 2 and all the following experiments, unless specified, 100 ng of rCP (approximate 3 pmol) or rE2 (as a control) were used per transfection.

GFP expression was detectable if the rCP was added 4 h before or after the transfection of RUBrep/GFP_ΔNotI RNA. If the RNA and rCP were added 24 h apart, no GFP was detected, showing that the window of CP accompanying RuV RNA is also important (data not shown).

### 2.3. RuV CP Does Not Affect Protein Synthesis

To investigate whether RuV CP increased the translation efficiency of RuV RNA, which might enhance replication, we assessed the translation efficiency from the RuV replicon RNAs in vitro using Flexi^®^ rabbit reticulocyte lysate with FluoroTect™ GreenLys labeling system (Promega, Madison, WI, USA) in the presence of rCP or rE2. The constructs included RUBrep/GFP (a full-length replicon), genomic mini-Xpress (g41-GFP or g1700-GFP), a subgenomic mini-Xpress (sg-GFP), and a control construct pCI-GFP, in which the GFP gene was expressed from the early promoter of human cytomegalovirus (CMV) (Promega) (Appendix A). A specific product of approximately 220 kDa was detected from RUBrep/GFP replicon (Appendix A). No differences were found in reactions with rCP or rE2. Similar observations were made using RuV g-GFP, sg-GFP, or GFP RNA from pCI-GFP with serially diluted RuV rCP or rE2 (Appendix A), indicating that RuV CP did not affect the translation from any of these RNA in vitro.

The effect of rCP on translation was also assessed in transfected cells with RuV mini-Xpress. The fluorescent proteins in g-GFP and sg-GFP transfected cells were detected within 4 h after adding the RNA. Supplementing rCP did not make a difference in the amount of GFP from the experiments with rE2 for up to 4 d post-transfection (Appendix A). The same results were observed using the control GFP RNA (data not shown). Similarly, in cells transfected with full-length RuV replicons expressing RFP-tagged NSP, supplementing rCP did not result in increases in fluorescence within 24 h post-transfection (Appendix A and see below). Taken together, these results indicate that exogenous RuV CP does not affect the translational efficiency of any tested messenger RNA, thus the enhanced replication by rCP is not due to the translation regulation from RuV ORFs.

### 2.4. Exogenous RuV CP Does Not Affect Intracellular Localization of RuV NSP

One proposed mechanism of intracellular CP-enhanced RuV replication was by chaperoning viral NSP to the endosomal membrane [29]. This was supported by the co-localization and interaction of RuV CP and NSP [5,29]. To investigate whether exogenous RuV rCP affected the intracellular distribution of RuV NSP, we used RuV replicons expressing RFP-tagged NSP (Figure 3A). Four hours after the addition of RNA, punctate red fluorescence scattered in the cytoplasm was detected (Figure 3B, far left images). In transfected cells with rCP, the distribution of these punctate fluorescent signals showed no difference from the transfection with rE2 between 4 and 18 h post-transfection. Moreover, similar intracellular localization of the fluorescent punctate was observed in transfected cells with non-replicating RuV RNA, such as g1700-Rfp (a genomic mini-Xpress expressing RFP) (data not shown) or RUBrep/GFP/NSP-Rfp_RdRp* (with the viral RdRp GDD domain mutated) (Figure 3B, far right images).

At 18 h, aggregates of the fluorescent punctate pattern were noticed. This shift of RFP distribution also occurred in cells with rE2, as well as in cells transfected with non-replicating replicons. Thus, this shift is apparently independent from the replication of replicon RNA (arrows in Figure 3B). At 48 h post-transfection, in cells with replicating replicons, intense, distinctive red fluorescent foci adjacent to the nuclei appeared (images at 72 h far left column in Figure 3C) and 5 d post-transfection (arrowhead, Figure 3B) are shown. These bright fluorescent foci were similar to the structures previously described in RuV infected cells [30].

Most of the perinuclear RFP foci were shown to colocalize with dsRNA using a mouse monoclonal antibody specific to double-stranded RNA (dsRNA) (J2; Scicons, Szirák, Hungary) (Figure 3C; GFP expressed from RUBrep/GFP was undetectable after methanol fixation (Appendix A). Thus, the green fluorescence represented the dsRNA instead of GFP fluorescence. No dsRNA signal was found in cells transfected with non-replicating replicons, even in the presence of RuV rCP. Supplemented RuV-recombinant SP was not detected in all these experiments by IFA (data not shown).

### 2.5. RuV CP Affects the gRNA, but Not sgRNA Synthesis

The data presented above indicated that RuV rCP enhances RuV replication before or during viral RNA synthesis. To closely examine which stage of viral RNA synthesis rCP is most effective, replication of RuV replicons known to have specific defects during genome replication was assessed in the presence of rCP or rE2. These replicons included RUBrep/GFP (WT), RUBrep/GFP_ΔNotI (or RUBrep/GFP/NSP-Rfp), RUBrep/GFP-1301S, and RUBrep/GFP-RdRp*. RUBrep/GFP-1301S is an NSP cleavage mutant with amino acid 1301 (a glycine) changed to serine. This change completely abolished the processing of NSP polyprotein precursor (p200). As a result, the synthesis of the (+) gRNA, but not the (−) RNA, was greatly reduced [31].

Replication of these replicons was first assessed by the dsRNA synthesis by IFA (Figure 4A). In cells transfected with replication-competent replicons, such as RUBrep/GFP and RUBrep/GFP_ΔNotI, significantly more dsRNA was detected in the presence of rCP, compared to the presence of rE2 (Figure 4A). Although replication of RUBrep/GFP_1301S could not be fully restored by rCP, as very few GFP-expressing cells were detected (Table 2), there were slight increases in the dsRNA signals in cells transfected with this replicon in the presence of rCP (Figure 4A). No dsRNA signal was detected in cells with RUBrep/GFP_RdRp*.

Consistent with the detection of dsRNA, (−) RNA was readily detected in the replication-competent replicons with rCP, while it was hardly seen with rE2 by Northern hybridization (Figure 4B; comparing lanes 1 to 5; lanes 2 to 6). The (−) RNA of RUBrep/GFP_1301S was only detected in the presence of rCP by more sensitive methods (Figure 4C). No (−) RNA was detected from RUBrep/GFP-RdRp*. The same patterns were also seen with (+) RNA by Northern hybridization (Figure 4B,C, bottom panel), except that (+) RNA was detected in all constructs by RT-PCR. Part of the amplification may have resulted from the transfected RNA transcripts.

rCP exerted marginal impact on sgRNA synthesis: from the replication of RUBrep/GFP, the ratio of (+) RNA to sgRNA was 0.86 ± 0.01 with rCP, and 0.77 ± 0.01 with rE2 at 3 to 6 d post-transfection from four independent experiments. To confirm this observation, we compared the replication of rubella replicons with the intergenic region (IR), which contained the putative sgRNA promoter [32] replaced with an internal ribosomal entry site (IRES) of encephalomyocarditis virus (EMCV) [33]. The expression of the reporter proteins from replicons with IRES therefore reflects the gRNA rather than the sgRNA synthesis (Figure 5A).

We used RuV replicons expressing puromycin *N*-acetyl-transferase (PAC), of which the replication capability can be measured by the number of puromycin-resistant cells such that the dependence on rCP can be quantifiable [11]. The reporter expression from RuV replicons with IRES was lower than the expression from the wild-type replicons (Figure 5B). This is due to the fact that in replicons with IRES, the reporter expression is from (+) gRNA rather than the sgRNA; the latter is present in a higher molar ratio in infected cells [34]. Replication of RUBrep/PAC and RUB-IRES-PAC, their respective constructs expressing RFP-tagged NSP, and a control pEXP-Lib (Takara Bio USA, Inc., San Jose, CA, USA), with or without rCP, were compared. More colonies were recovered from puromycin selection if rCP was supplemented during transfection (Figure 5B): the ratio of antibiotic-resistant cells with and without rCP after transfection with RUBrep/PAC was 2.7, while this ratio was 6.4 after the transfection with RUB-IRES-PAC. The rCP/rE2 ratio of antibiotic-resistant cells in pEXP-Lib transfection was 0.66 ± 0.11 (Figure 5C). In the transfection with RUBrep/PAC/NSP-Rfp or RUB-IRES-PAC/NSP-Rfp, very few cells survived antibiotics without rCP (Figure 5B). Once the replication of PAC replicons is established, the drug-resistant cells can be maintained without rCP (data not shown).

### 2.6. RuV CP Rescues the Replication of a Spectrum of RuV Mutants and Affects RNA Stability

The results from above showed that, similar to the de novo synthesized intracellular CP [12], rCP mainly enhanced replication of the gRNA, particularly (−) RNA synthesis (Figure 4 and Figure 5), suggesting that CP have a specific function in copying the genomic RNA (possibly being a part of the replication complex) and/or improving the intracellular machinery that favors viral replication, e.g., an environment which leads to less viral RNA degradation. By screening a spectrum of replicons with mutations in the termini [9,12] and NSP, we noticed that rCP rescued some but not all of the mutants, and some mutants were rescued when both wild-type NSP and rCP were present (Table 2). The mutants that were rescued by rCP alone had defects in p150 or p90; in addition to the replicons with deletions in the Q domain (ΔNotI or NSP-RFP), replicons with a mutation in the NSP cleavage site (1301S), in the RdRp domain (GA1967D), or with extended C-terminus (p90-His) were all rescued by rCP. Since the mutations rescued were in many locations, the possible mechanism of rCP-enhanced replication by trans-complementing a specific domain in the replication proteins is unlikely. To test if rCP enhanced replication by increasing RNA stability, we first monitored the intracellular fluorescence of fluorescently labeled RUBrep/Rfp_RdRp* and pCI-Rfp RNA over a 5 d timeframe. The total fluorescence from each time point was normalized to the fluorescence at day 0. We found that, at 24 h post-transfection, there were approximate 1.33-fold more signals from RuV RNA species if the RNA was mixed with rCP than that with rE2, while this ratio was ~1.02 with control RNA (Figure 6A). To confirm this observation, we compared the effect of rCP on RuV RNA retention from different constructs, including RUBrep/GFP (WT), RUBrep/GFP_NSP-ATG* (ATG*; a replicon with NSP start codon mutated to TAG), RUBrep/GFP_RdRp*, and RNA from pCI-GFP, in cells within 24 h post-transfection. The changes in the amount of target RNA over a period of time was calculated using the fold change expression method (ΔΔCt), where the ΔCt (=Ct_GFP_ − Ct_GAPDH_) at the specific time point was subtracted from the ΔCt at 0 h time point. The relative ΔΔCt (rΔΔCt = ΔΔCt_rCP_*N*-h_/ΔΔCt_rE2_*N*-h_), which was calculated by the ratio of the ΔΔCt from the transfection with rCP and the ΔΔCt from the transfection with rE2, was used to reflect the effect(s) (or dependence) of rCP on the input RNA template, in comparison with the presence of rE2, over a period of time.

At 6 h post-transfection, the rΔΔCt of three RuV replicons were within the range of 1.19–1.38 without any significant difference (*p* > 0.8). On the other hand, the rΔΔCt of non-RuV RNA was 0.73 ± 0.14 (Figure 6B). At 18-24 h post-transfection, the rΔΔCt of replication-competent replicon (WT) was 2.25 ± 0.98 (Figure 6B, data not shown), whereas the rΔΔCt of non-replicating replicons, ATG* and RdRp*, and pCI-GFP control RNA exhibited no significant difference from the rΔΔCt at 6 h (*p* > 0.5) with values of 1.17 ± 0.17, 1.12 ± 0.19, and 0.67 ± 0.17 respectively. These are statistically significant compared to the rΔΔCt of control pCI-GFP RNA (*p* < 0.05). Taken together, these results indicate that the exogenous rCP retained input RuV RNA marginally but steadily within 24 h after entry.

## 3. Discussion

The exact mechanism of how de novo synthesized CP enhance genome replication is not clear, but several observations have been reported. These include modulating gRNA synthesis [12] and upregulating sgRNA synthesis [35]. In addition, de novo synthesized CP were also found to interact with RuV NSP [15,29] and several host factors [36,37,38,39]. Nevertheless, the roles of CP in virions on RuV genome replication have not been well-recognized. It is possible that the exogenous CP enhancement of RNA replication could have early functions that are different from those of the de novo synthesized CP later. For example, different phosphorylation states could result in differential affinity to viral RNA [17,18] and to host factor Bax [39]. The enhancement by exogenous CP was found to be specific for RuV RNA replication. Indeed, reduced activities in non-RuV virus or reporters were noticed throughout the entire study, suggesting this particular property of RuV antigens mainly targets RNA intracellularly; otherwise, there should be limited impact on the non-RuV RNA species. The enhancement was not an outcome of a general increase in transfection efficiency. We examined the early stages of genome replication that CP might have influence over and found that CP has more impact on the gRNA synthesis, especially (−) RNA, at least partially through physical interaction with RuV RNA. Thus, a novel role of extracellular RuV CP on genome replication early in the virus replication cycle (i.e., immediately after disassembly) is suggested.

De novo synthesized CP has been reported to inhibit protein synthesis [37] and cellular protein importation into the mitochondria [39]. RuV CP was also linked to apoptosis, although the role of CP on apoptosis is controversial [39,40,41]. We did not observe any impact on the translation from the NSP or SP ORF at the concentration of rCP used in this study. The temporal requirement of exogenous CP to enhance RuV genome replication suggests that productive replication can be achieved if viral RNA is accompanied by exogenous CP within a specific window during the early stage of the replication cycle. The presence of exogenous CP did not affect the intracellular distribution of NSP prior to viral RNA synthesis and dramatic differences were not detected till active (−) gRNA synthesis began [31]. Although our attempts to track rCP were unsuccessful, the fact that rCP is not needed to maintain cells expressing RuV PAC replicons clearly indicates that effect of exogenous CP on genome replication is transient. It is not clear whether the interaction of host factors with exogenous CP engages in viral genome replication directly (e.g., trans-localization of NSP). Investigations on the host responses prior to viral RNA replication should elucidate the involvement of cellular proteins in the early stage of viral replication.

Knowing the detailed actions of CP in replication is challenging since CP is not required for replication [9]. Involvement of de novo synthesized CP in viral genomic RNA synthesis as well as NSP maturation were reported [12,14,29]. Although CP helped with establishing and/or stabilizing replication, it is less likely to complex directly with the RNA replication machinery; CP is less likely to complement specific functions of NSP [13,14] since it restores the replication of mutants with modifications across different domains in the NSP (this study; [29]) and termini. We favor the simpler explanation that exogenous CP has an early effect in the early post-entry replication [42] by direct interaction with viral RNA and affects genomic RNA synthesis at or prior to (−) RNA synthesis. Retention of RuV RNA was observed among all RuV constructs, including those with severe defects in the replication or translation from the NSP ORF, the latter confirming that the enhancement was independent from the translation machinery. While it becomes more prominent among replication-competent replicons as robust viral RNA replication progresses [31], the non-replicating RNAs were steadily maintained. The impact of exogenous CP at an earlier time point, although marginal, may be sufficient for viral RNA to proceed to the next stage of the replication cycle, i.e., the (−) RNA synthesis. This also explains why no significant difference was observed in the translation efficiency from the NSP ORF of the replicon RNA with or without rCP within 24 h post-transfection. Based on the data we presented, we propose that exogenous CP catapult viral genome replication during the earliest stage of the viral replication cycle by protecting viral RNA, for example, and facilitating the achievement of the equilibrium between viral replication (e.g., translation from the SP ORF) and host response [43]. Other mechanisms such as facilitating the organization of replication sites, the organization of RuV RNA into a conformation that is effectively replicated (an RNA chaperon), and/or p200 maturation [29] also cannot be ruled out. To our knowledge, this is the first study providing direct evidence that the exogenous viral nucleocapsid proteins can be involved in viral genome replication. Speculation of enhanced genome replication by exogenous CP has been drawn [29,42]; however, it was based on the observation from the uses of package cell lines and RuV replicons. Under such conditions, recombination between RuV replicon RNA and SP can happen, and as a result infectious virus was generated [44]. The mechanism on how RuV rCP was internalized by cationic liposomes is not clear, but this approach has been adapted in the introduction of Cas9 endonuclease [45]. The findings of enhanced viral replication by exogenous RuV CP may have some practical value in clinical diagnosis. For example, supplementation of RuV CP or virion proteins may expedite recovery of infectious virus from a small amount of RuV RNA. Occasionally, when no infectious virus is recovered due to suboptimal storage/transport conditions of clinical specimens, adding RuV CP with RNA extracted from patients’ samples may improve the likelihood of RuV isolation.

## 4. Materials and Methods

### 4.1. Cell Culture, Virus Preparation, and Titration

The BHK and Vero cell lines were obtained from ATCC (Manassas, VA, USA). The BHK clonal cell lines expressing RuV SP were described previously [12]. RuV infectious cDNA clones (Robo402 and Robo402ires) and RuV replicons (RUBrep/GFP and RUBrep/GFP_ΔNotI) were obtained from Dr. Teryl Frey from Georgia State University.

BHK (ATCC, Manassas, VA, USA) or BHK clonal cells expressing RuV proteins or Vero cells (ATCC) were maintained in Dulbecco’s Minimal Essential Medium (DMEM) (Thermo Fisher Scientific, Waltham, MA, USA) supplemented with 5% fetal bovine serum (FBS) (Thermo Fisher Scientific) and gentamicin (10 mg/mL; Thermo Fisher Scientific) at 37 °C and 5% CO_2_. When maintaining BHK clonal cells, the medium also contained 0.8 mg/mL geneticin (Thermo Fisher Scientific).

RuV infection and immunofluorescent assays (IFA) were performed as previously described [11,46]. Fluorescent conjugate secondary antibodies and nuclear stains (DAPI and Hoechst 33258) were from Thermo Fisher Scientific. A mouse monoclonal antibody to double-stranded RNA (J2) was purchased from Scicons (Szirák, Hungary). Virus quantitation was performed using an immunocolorimetric assay (ICA) [47]. Briefly, serially diluted virus was used to infect Vero cells in 48-well plates. After infection, the cells were overlaid with agar-media mix and incubated for 4 d in a 37 °C incubator with 5% CO_2_. The titer of the virus was determined by the number of foci stained by ICA using an in-house mouse monoclonal antibody to RuV E1 glycoprotein [47].

Sindbis virus (SINV) infection was carried out in BHK-21 cells. The SINV stock was collected at 24 h post-infection and titrated by plaque assay in BHK cells. UV-inactivation of SINV (10^6^ pfu/mL) was done as described previously [48].

### 4.2. Virus Antigens and Pseudovirus

Preparation of non-infectious viral antigens was carried out by repeated freeze-thaws of a commercial UV-inactivated virus stock (Meridian Life, Memphis, TN, USA). Successful inactivation was confirmed by the absence of RuV replication (i.e., no de novo synthesis of RuV E1) in cells inoculated with the inactivated stock and in a subsequent passage (P1) by ICA. Previously described RT-PCR using intracellular RNA from the first passage was also used to confirm the absence of infectious virus [27]. Rubella virus-like particles (VLP) were purchased from Advanced ImmunoChemical (Long Beach, CA, USA). Per the manufacturer’s information, the material consisted of rubella SP of F-therien strain and was produced in HEK293 cells. Both UV-inactivated virus and RuV VLP are referred to as RuV antigens in this text. Recombinant RuV CP (rCP; full-length rubella capsid proteins with a His-tag at C-terminus from Saccharomyces cerevisiae per the manufacturer’s information) and E2 proteins (rE2; bacterially expressed recombinant protein encompassing aa 31–105 per the manufacturer’s information) were purchased from Abcam (Cambridge, MA, USA). Lyophilized commercial reagents were resuspended in serum-free Opti-MEM (Thermo Fisher Scientific) prior to use.

RuV pseudovirus was prepared by transfecting BHK clonal cells expressing RuV SP [12] with RuV replicon RNA transcribed in vitro using Lipofectamine 2000 (Thermo Fisher Scientific). At the end of transfection, the Lipofectamine 2000-RNA mix was removed, and cells were overlaid with DMEM supplemented with 2% FBS followed by incubation in a 37 °C, 5% CO_2_ incubator. The pseudovirus stock (P0) was prepared by collecting the growth media from transfected cells every 24 h post-transfection up to 7 d post-transfection. Prior to inoculation of Vero cells, culture media collected from transfected BHK cells was centrifuged at 2000× *g* and approximately 1/50 volume of the supernatant was used for inoculation. The absence of infectious virus in the pseudovirus stock was confirmed by ICA and RT-PCR as described above.

### 4.3. Constructs and Transfection

Table 3 lists all constructs used in this study. The constructs were either made by the standard site-directed mutagenesis or using an asymmetric PCR strategy [46]. The cleaning, digestion, and ligation of PCR fragments to appropriate expression vectors by molecular cloning was done as described in [9]. A Sabin 2 poliovirus (PV) molecular clone was transcribed in vitro to generate infectious full-length viral RNA as described previously [28]. Sabin 2 was used in accordance with the GAPIII regulations.

Transfection was carried out using Lipofectamine 2000 as per the manufacturer’s protocol. Unless specified, all transfections were carried out using Vero cells in 48-well plates. Briefly, 1–2 μg of RNA, determined by spectrophotometry [NanoDrop ND-1000 spectrophotometer (Thermo Fisher Scientific)], were mixed with Lipofectamine 2000 in Opti-MEM followed by inoculation onto Vero cells for 4 h in a 37 °C, CO_2_ incubator. In experiments that required dilution of RNA, the RNA was serially diluted with yeast tRNA (Roche, Basel, Switzerland) to ensure that the same amount of RNA was used in transfection. In transfection supplemented with RuV virion proteins, unless specified, 100 ng of UV-inactivated virus, VLP, or recombinant RuV structural proteins diluted in Opti-MEM were added with RuV replicon RNA-Lipofectamine 2000 mix prior to the addition onto Vero cells.

Transfection with RuV replicons encoding a puromycin-resistant gene was carried out using BHK cells in 96-well plates. Cells were trypsinized and seeded in 6-well plates at the end of transfection, grown in DMEM with 5% FBS, and subjected to puromycin selection (2 μg/mL) at 24 h post-transfection. The numbers of surviving cells were counted using a hemocytometer and the cell colonies were subjected to crystal violet staining at 6 to 7 d post-transfection.

### 4.4. In Vitro Transcription and Translation

Synthesis of capped RNA in vitro was accomplished with SP6 or T7 RNA polymerase (New England Biolabs, Ipswich, MA, USA), and m^7^G(5′)ppp(5′)G RNA cap analog (New England Biolabs) as described previously [9]. Synthetic RNA incorporating fluorescent or DIG labels were made using ribonucleotides mixed with ChromaTide^®^ Alexa Fluor^®^ 488-5-UTP- (Thermo Fisher Scientific) or Digoxigenin-11-UTP- (Sigma-Aldrich, Burlington, MA, USA) during in vitro transcription as per manufacturers’ protocol. The free labels were removed by Sephadex G-50 quick spin columns (Sigma-Aldrich).

In vitro translation was carried out using the Flexi^®^ Rabbit reticulocyte lysate and the nonradioactive FluoroTect™ Green_Lys_ in vitro translation labeling as per the manufacturer’s protocol (Promega, Madison, WI, USA). In brief, approximately 100–200 ng RNA transcripts were used in a 15 μL-reaction with indicated recombinant RuV proteins. The results were resolved on 10 or 12% SDS-polyacrylamide gels and visualized using a Typhoon 9400 Imager (GE Healthcare Life Sciences, Pittsburgh, PA, USA). Data analysis was accomplished with ImageQuant software (GE Healthcare Life Sciences).

### 4.5. Northern Hybridization and RT-PCR

Northern hybridization to detect (−) and (+) RNA from transfected cells using NorthernMax™-Gly hybridization kit (Thermo Fisher Scientific) was described previously [12]. In brief, intracellular RNA was collected using Tri-reagent RT (Molecular Research Center; Cincinnati, OH, USA), which allowed separation of RNA from DNA contaminants. Total RNA was resuspended in 80% DMSO. Prior to loading onto 0.85% agarose gel, RNA was denatured at 80 °C for 15 min then mixed with an equal volume of NorthernMax-Gly sample loading dye. Detection of strand-specific RNA species was accomplished using DIG-UTP labeled RNA probes comprised of approximately 3′ 1300 nts of RUBrep/GFP. In general, four-fifths of the total RNA was used for the detection of (−) RNA and one-fifth of the total RNA was used for the detection of (+) RNA.

To detect (−) RNA and (+) RNA by RT-PCR, aliquoted total RNA was annealed with 1 μM strand-specific primers: to detect (−) RNA, GFP (+) (5′-ATG GTG AGC AAG GGC GAG CTG TTC-3′) was used; while to detect (+) RNA, GFP (−) (5′-TTA CTT GTA CAG CTC GTC CAT GCC GTG AGT-3′) was used. The final product was approximately 700 base pairs (bp). In brief, total RNA extracted from transfected cells in 80% DMSO was denatured at 80 °C for 15 min and annealed to the strand-specific primer followed by ethanol precipitation. The probe-RNA was resuspended in nuclease-free water and subjected to reverse transcription (RT) using SuperScript II reverse transcriptase (Thermo Fisher Scientific). After reverse transcription, reactions were treated with 500 ng DNase-free RNase (Sigma-Aldrich) and cleaned with Agencourt AMPure XP module at a ratio of 1:1 (Beckman Coulter Genomics, Chaska, MN, USA). The cDNA was eluted in 50 uL nuclease-free water and amplified with GFP (+) and GFP (−) primers and Ex-taq DNA polymerase (Takara Bio USA, Inc., San Jose, CA, USA). The PCR products were resolved on 1.5% agarose gels.

### 4.6. Quantification of Reporter Proteins and RNA

The intracellular fluorescence was measured using a FluoroSkan Ascent Microplate fluorometer (Thermo Fisher Scientific). Cell monolayers were washed once with Opti-MEM and stained with 2.5 μg/mL Hoechst 33258 (Thermo Fisher Scientific) in a 37 °C, 5% CO_2_ incubator for 30 min. After incubation, the monolayers were washed thrice with PBS and lysed in a buffer containing 100 mM Tris-HCl (pH 7.5) and 0.05% SDS. The total fluorescence was measured from the cell lysate in 96-well black plate with filter sets of 485 nm (excitation) and 527 nm (emission) for green fluorescence (ChromaTide^®^ Alexa Fluor^®^ 488-5-UTP-labeled RNA and GFP), and 355 nm (excitation) and 405 nm (emission) for Hoechst 33258 staining. By this means, a linear decline in fluorescence was noted with serially diluted fluorescently labeled RNA by a fluorometer. The relative fluorescence was calculated by taking the ratio between green fluorescence and Hoechst 33258 (F_G_Day·*n*_/F_H_Day·*n*_; where *n* indicates days post-transfection) at the *n*th day post-transfection compared to the ratio taken at the day 0 time point [(F_G_Day·N_/F_H_Day·N_)/(F_G_Day 0_/F_H_Day·0_)].

To assess β-galactosidase in pCH110-transfected cells (GE Healthcare Life Sciences), cellular proteins from 5 d post-transfected cells were collected following three freeze-thaw cycles in PBS. The amount of intracellular β-galactosidase was determined from approximately one-third of the lysate using the Luminescent β-galactosidase detection kit II (Takara Bio USA, Inc.) and was analyzed by a TD-20/20 Luminometer (Turner Biosystem; Sunnyvale, CA, USA).

Quantification of RuV RNA by Northern hybridization was carried out by densitometry scanning of the chemiluminescent signal after exposures on X-ray films by UVP BioImaging Systems. The quantification was analyzed using LabWorks 4.0 Image Acquisition and Analysis software (UVP LLC; Upland, CA, USA).

Intracellular RNA retention was determined using the comparative CT method (ΔΔCt). In vitro transcripts were treated with two cycles of DNase I (Promega) followed by cleaning using an RNAeasy kit (Qiagen, Hilden, Germany) prior to transfection. The GFP RNA was detected with primers and probe specific to the GFP reporter gene (forward primer: 5′-ATC ATG GCC GAC AAG CAG AAG AAC-3′; reverse primer: 5′-GTA CAG CTC GTC CAT GCC GTG AGT-3′; probe: 5′-FAM-CAG GAC CAT GTG ATC GCG CTT CTC GT-BHQ-3′) [49]. The housekeeping glyceraldehyde-3-phosphate dehydrogenase (GAPDH) gene was chosen as a reference gene with primers (forward: 5′-GAA GGT GAA GGT CGG AGT C-3′; reverse primer: 5′-GAA GAT GGT GAT GGG ATT TC-3′) and probe (5′-Cy5-CAA GCT TCC CGT TCT CAG CC-BHQ2-3′) [50] using the QuantiFast Multiplex RT-PCR Kit (Qiagen). Thermal cycling was carried out with an ABI Prism 7500 Real-Time PCR System (Thermo Fisher Scientific) and the data were analyzed with SDS software (version 2.0.6, Applied Biosystems, Waltham, MA, USA) (Thermo Fisher Scientific). The fold expression of GFP gene at any time point of post-transfection was measured by normalizing to GAPDH expression using the fold change expression method (2^−^^ΔΔCT^). The ΔCt at a later time point *n* was used to compare with the ΔCt value at 0 h (ΔCt_0_) to obtain the difference, ΔΔCt*_n_*. The ΔΔCt*_n_* was plugged into the equation of 2^−^^ΔΔ^^Ct^ to calculate the change in the amount of target RNA over a period of time. The relative ΔΔCt (rΔΔCt*_n_* = 2^−^^ΔΔ^^Ct_rCP^/2^−^^ΔΔ^^Ct_rE2^) was used to indicate the ratio of the amount of input RNA in the presence of rCP to the amount of RNA in the presence of rE2 at the specific time point. Quantification of PV RNA was performed using a real-time RT-PCR assay for the 3Dpol coding region, and PV titer was determined by standard plaque assay as described previously [48].

### 4.7. Data Analysis

Unless specified, the effects of rCP on the reporter gene expression, amount of RNA, or antibiotic resistant cells were presented by the quantitative comparison using the data from experiments with supplemented rCP to the experiments with supplemented rE2 (rCP/rE2). All data were graphed using Microsoft Excel or GraphPad Prism v.9. Kolmogorov–Smirnov test and Student’s T-test were performed using GraphPad Prism^®^ v.9 (GraphPad Software, San Diego, CA, USA) from at least three independent experiments. A statistical value *p* < 0.05 was considered significant.

## Figures and Tables

**Figure 1 pathogens-11-00683-f001:**
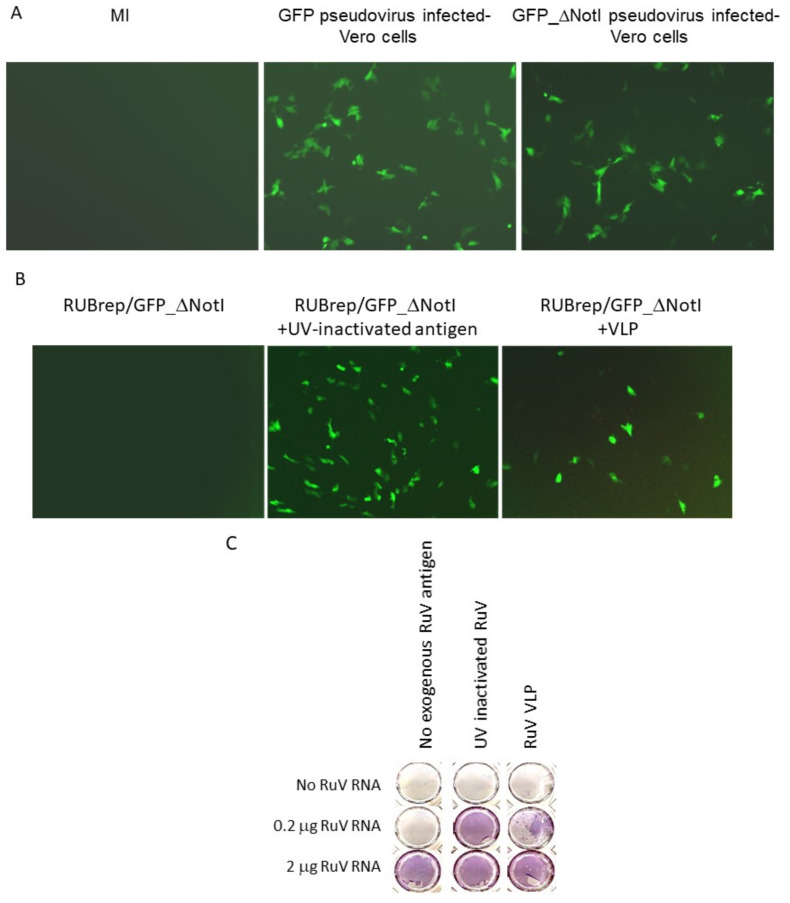
Enhanced replication of RUBrep/GFP_ΔNotI replicons. The images in (**A**,**B**) were taken at 2 d post-transfection using the Axiovert 200 (Carl Zeiss, Göttingen, Germany) at 100×. (**A**) GFP expression in cultures infected with RUBrep/GFP or RUBrep/GFP_ΔNotI–pseudovirus, as indicated at the top of the images. MI: uninfected cells. (**B**) GFP expression in cells transfected with RUBrep/GFP_ΔNotI RNA (1–2 μg) with or without RuV antigens (100 ng per transfection) as indicated, including UV-inactivated RuV antigens and RuV VLP. (**C**) Recovery of RuV from infectious RNA by exogenous RuV antigens. Run-off transcripts of RuV RNA from an RuV-infectious cDNA clones were transfected into Vero cells in 48-well plates with (column 2: 100 ng of UV-inactivated RuV antigens; column 3: 100 ng of VLP) or without RuV antigens (column 1). The amount of RuV RNA in each transfection is indicated on the left side of the panel. Cells were fixed and stained by ICA at 5 d post-transfection using an in-house mouse monoclonal antibody to RuV E1 glycoprotein.

**Figure 2 pathogens-11-00683-f002:**
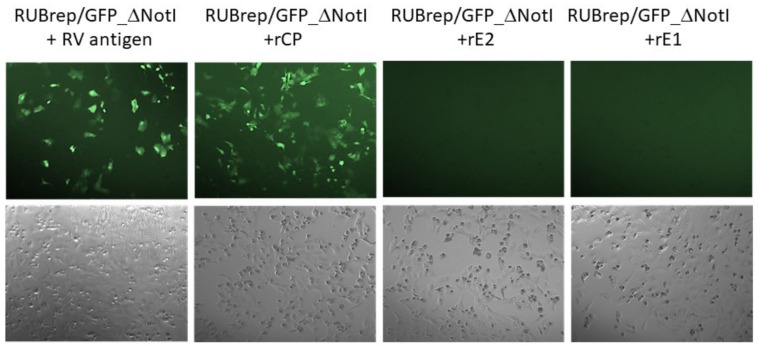
GFP expression in RUBrep/GFP_ΔNotI-transfected Vero cells with supplemented recombinant RuV proteins. Vero cells in a 48-well plate were transfected with the replicon RNA mixed with 100 ng of inactivated RuV antigen or recombinant RuV SP (Abcam, Cambridge, MA, USA) as indicated. The images were taken at 48 h post-transfection at 100× by Axiovert 200 (Carl Zeiss). The phase contrast images for each of the four transfections taken from the same field are shown at the bottom.

**Figure 3 pathogens-11-00683-f003:**
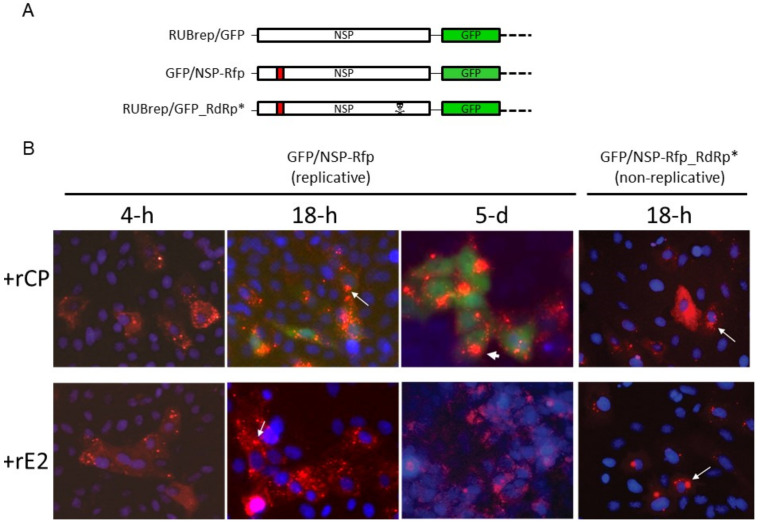
Intracellular localization of RuV macromolecules in replicon-transfected cells. (**A**) Schematic representation of RUBrep/GFP, RUBrep/GFP/NSP-Rfp (“NSP-Rfp”), and RUBrep/GFP/NSP-Rfp_RdRp* (NSP-Rfp_RdRp*). (**B**) GFP and RFP expression in transfected cells with RuV replicons with rCP (top) or rE2 (bottom). The live images were taken at the indicated time point of post-transfection at 200×. Cell nuclei were stained with Hoechst 33258 (Thermo Fisher Scientific, Waltham, MA, USA) and are shown in blue fluorescence. (**C**) Colocalization of RuV NSP and dsRNA. Cells transfected with RUBrep/GFP/NSP-Rfp or RUBrep/GFP/NSP-Rfp_RdRp* with supplemented rCP or rE2 were fixed at 3 d post-transfection with methanol and subjected to staining using dsRNA specific mouse monoclonal antibody J2 (Scicons, Szirák, Hungary) at 1 to 2000 dilution followed by incubation with Alexa Fluor^®^ 488-conjuagted goat-anti-mouse IgG (Thermo Fisher Scientific). Cell nuclei were stained with DAPI. Images at 100× were taken with individual filter channel or combined channels (merge) as indicated. Red: NSP; green: dsRNA; blue: nuclei.

**Figure 4 pathogens-11-00683-f004:**
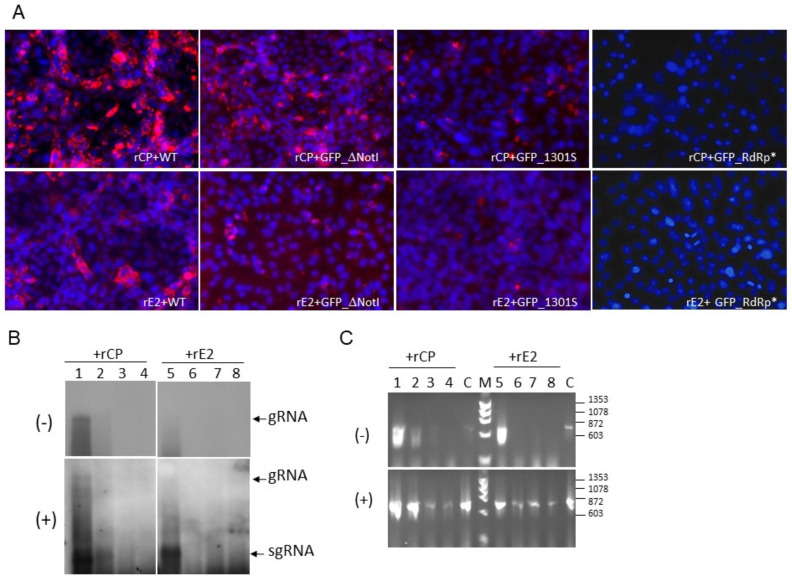
RuV RNA synthesis in the presence of rCP or rE2. RuV replicons, RUBrep/GFP (WT), RUBrep_ΔNotI (GFP_ΔNotI), RUBrep/GFP_1301S (GFP_1301S), and RUBrep/GFP_RdRp* (GFP_RdRp*) were transfected to Vero cells with supplemented rCP or rE2. Cells were fixed or the RNA was harvested at 3 d post-transfection. (**A**) Detection of dsRNA (red fluorescence) in replicon-transfected cells as indicated at the bottom of each image with RuV rCP (top images) or rE2 (bottom images) by IFA using J2 mouse monoclonal antibody and a goat-anti-mouse IgG secondary antibody conjugated with Alexa Fluor^®^ 546 (Thermo Fisher Scientific). Cell nuclei were stained by DAPI (blue fluorescence). The images were taken at 100×. (**B**,**C**) Detection of strand-specific RuV RNA in transfected cells by Northern hybridization (**B**) or RT-PCR (**C**). Strand-specific RNA species, as indicated at the left of the blots, were detected using RNA probes of specific polarity by Northern hybridization. (**C**) Detection of RuV RNA species by RT-PCR. The sizes of molecular marker (M) are shown at the right (Molecular Weight Marker IX; Roche, Basel, Switzerland). In both (**B**,**C**), lanes 1–4 were from transfection with supplemented rCP while lanes 5–8 were from transfection with supplemented rE2; lanes 1 and 5: RUBrep/GFP; lanes 2 and 6: RUBrep/GFP/NSP-Rfp; lanes 3 and 7: RUBrep/GFP/NSP-Rfp_1301S; lanes 4 and 8: RUBrep/GFP/NSP-Rfp_RdRp*. gRNA: genomic RNA; sgRNA: subgenomic RNA. C: RT-PCR reaction controls using GFP control RNA of specific strand polarity transcribed from pGEM-GFP.

**Figure 5 pathogens-11-00683-f005:**
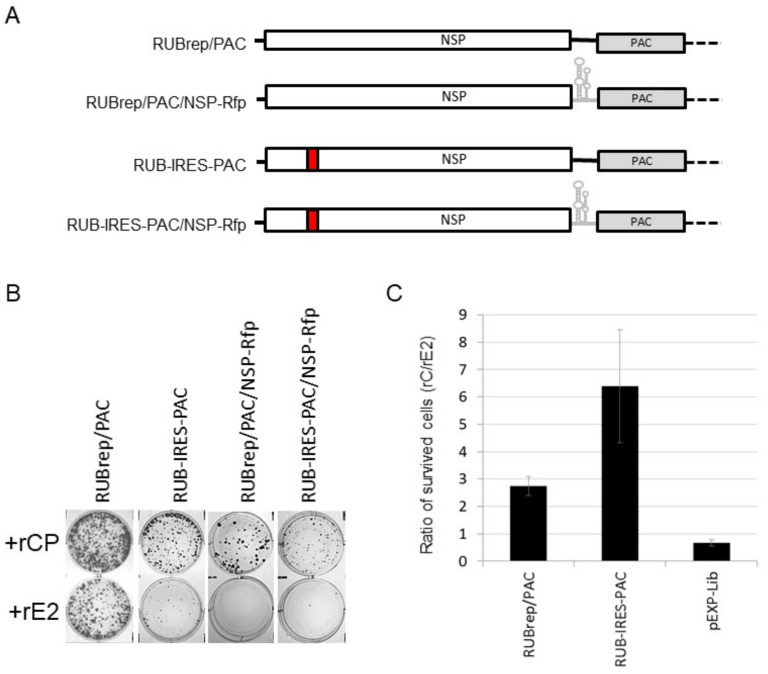
Effect of exogenous CP on the replication of IRES-containing replicons. (**A**) Schematic representation of IRES-containing replicons derived from RUBrep/PAC. (**B**) BHK cells survived puromycin selection after transfection with the RuV PAC replicons, as indicated, in the presence of rCP (top) or rE2 (bottom). Cells after transfection were seeded onto 6-well plates and subjected to puromycin selection at 2 μg/mL at 24 h post-transfection and stained with crystal violet at 7 to 10 d post-transfection. (**C**) Puromycin resistance by RuV PAC replicons or pEXE-Lib with rCP or rE2. BHK cells were transfected with RuV PAC replicons with rCP or rE2, subjected to puromycin selection and cells survived antibiotics were counted at 7 d post-transfection. The plot is taken from the ratios of puromycin-resistant cells from the transfection with rCP to the number of cells from the transfection with rE2 from four independent experiments.

**Figure 6 pathogens-11-00683-f006:**
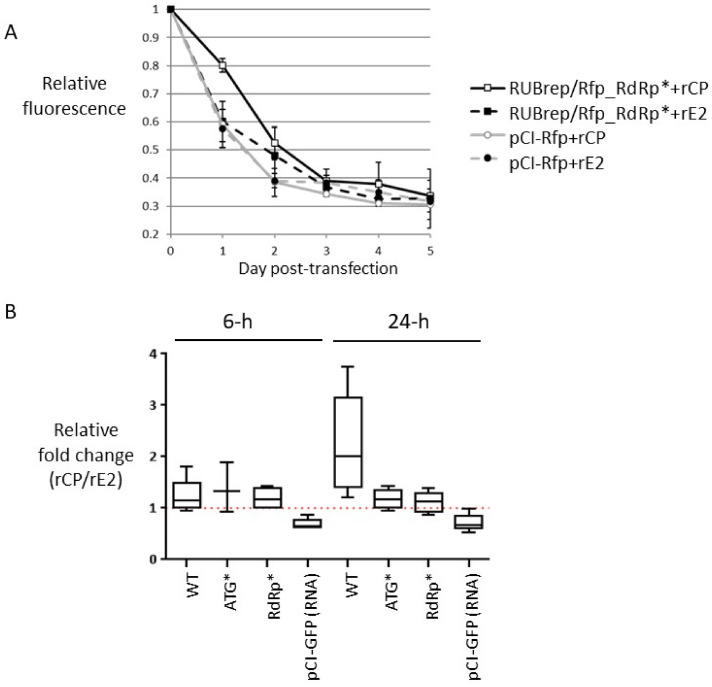
Retention of RNA in the transfection supplemented with rCP or rE2 by fluorometry (**A**) or real-time RT-qPCR (**B**). (**A**) Dynamics of intracellular fluorescence from fluorescently labeled RNA in the presence of rCP or rE2 over 5 d in transfected cells. Prior to the harvest, cells were stained with Hoechst 33258 followed by three washes with PBS. The intracellular fluorescence was measured using a Fluoroskan Ascent Microplate Fluorometer (Thermo Fisher Scientific) and the relative fluorescence was plotted on the Y axis against the day of post-transfection (X axis). Standard deviation was taken from three independent experiments. (**B**) Effect of rCP on the stability of transfected RNA within 24 h post-transfection. In vitro transcribed RNA from RuV replicons as indicated (X axis), were transfected Vero cells with rCP or rE2. Cell monolayers were washed thrice at the end of transfection to remove excess materials. Intracellular RNA was collected at 0, 6, and between 18 and 24 h post-transfection. The fold change expression of GFP gene at each time point was measured by normalizing to the expression of GAPDH. The graph represents the ratio (rΔΔCt) of the measurement from at least three independent experiments.

**Table 1 pathogens-11-00683-t001:** Specificity of rubella VLP on genome replication or gene expression ^1^.

	SINV	PV	RuV	Lac*Z* Control
Fold change in titer or activity (+VLP/−VLP)	0.14 ± 0.08	0.08 ± 0.06	2.99 ± 1.62	0.91 ± 0.28

^1^: Standard deviation was obtained from at least three independent experiments.

**Table 2 pathogens-11-00683-t002:** GFP expression from RuV replicons with mutations in NSP ^1^.

Mutant	Description	Domain	GFP Expression ^a^		References
			+rCP	+rE2	
ATG*	Mutate NSP ORF start codon to TAG	−	−	−	This study
ΔNotI	Delete nt 1694–2191 (aa 552–717) from NSP	In P150; within Q domain	++	−	[10,14]
NSP-Rfp	Replace nt 1694–2191 with RFP gene	In P150; within Q domain	++	−	This study
1152S	Mutate Cys1152 to Ser	In P150; catalytic domain of viral protease	+ ^2^	−	[31]
1301S	Mutate Gly1301 to Ser	Cleavage site by RuV protease	+	−	[31]
GA205D	Mutate Asp 205 to Ala	In p150; unknown domain	−	−	This study
GA1326D	Mutate Asp 1326 to Ala	In P90; unknown	+ ^2^	−	This study
GA1967D	Mutate Asp 1967 to Ala	In p90; putative RdRp catalytic domain	++	−	[6]
RdRp* (GK1967L1968)	Mutate Asp 1967 to Lys and Asp 1968 to Leu	In P90; putative RdRp catalytic domain	−	−	This study
P90-His	Add six histidine (His) residues at the C-terminus of p90	In p90; unknown	++	−	This study

^a^: +: Very few GFP-expressing cells were found in the entire lawn; ++: GFP-expressing cells were easily spotted; −: no GFP-expressing cells were detected. ^1^: Schematic representation of some constructs is shown in Appendix A. ^2^: GFP was only detected if functional NSP (from co-transfected RUBrep/Rfp RNA) was supplemented.

**Table 3 pathogens-11-00683-t003:** Constructs and cloning methods used in this study.

	Constructs	Description	Mutagenesis	Refs.
RuV infectious clone	Robo402	RuV infectious cDNA clone	ND (not needed)	[9]
	Robo402ires	RuV infectious cDNA clone with the intergenic region replaced with an internal ribosomal entry site (IRES) of encephalomyocarditis virus (EMCV)	ND	
RuV replicons	RUBrep/GFP	RuV replicon with partial SP coding region replaced by green fluorescent protein (GFP) gene	ND	[10]
	RUBrep/Rfp	RuV replicon with partial SP coding region (nt 6512 to 9333) replaced by red fluorescent protein (RFP) gene	PCR amplified RFP gene and swapped with the Xba I-Nsi I fragment in RUBrep/GFP	
	RUBrep/PAC	RuV replicon with partial SP coding region (nt 6512 to 9179) replaced by puromycin-N-acetyltransferase (PAC)	ND	[12]
	RUBrep/GFP_ΔNotI	RUBrep/GFP with nt 1693 to 2191 deleted	ND	[10]
	RUBrep/GFP/NSP-Rfp	RUBrep/GFP with RFP gene inserted between nt 1693 and 2191	PCR amplified RFP gene and swapped with the Not I region in RUBrep/GFP	
	RUBrep/PAC/NSP-Rfp	RUBrep/PAC with RFP gene inserted between nt 1693 and 2191	PCR amplified RFP gene and swapped with the Not I region in RUBrep/PAC	
	RUB-IRES-PAC	RUBrep/PAC replicon with the intergenic region (nt 6392–6511; IR) replaced by the EMCV IRES	PCR amplified partial NSP and IRES element of Robo402ires and swapped with the Fse I and Xba I fragment in RUBrep/PAC	
	RUB-IRES-PAC/NSP-Rfp	RUBrep/PAC/NSP-Rfp replicon with IR replaced by EMCV IRES element	PCR amplified partial NSP and IRES element of Robo402ires and swapped with the Fse I and Xba I fragment in RUBrep/PAC/NSP-Rfp	
	RUBrep/GFP_NSP-ATG*	RUBrep/GFP replicons with changes in the start codon ATG to TAG	PCR amplification with mutagenic primers and swapped the HindIII-Bsu36I fragment (nt 1–499)	
	RUBrep/GFP_1152S	RUBrep/GFP replicons with a single mutation in the catalytic pocket of RuV nonstructural protease at Cys1152 to Ser	PCR amplification with mutagenic primers and swapped the Bsu36 II-Cla I fragment (nt 499–4392)	[31]
	RUBrep/GFP_1301S	RUBrep/GFP replicons with the cleavage site (Gly 1301) of nonstructural polyprotein mutated to Ser	PCR amplification with mutagenic primers and swapped the Bsu36 II-Cla I fragment (nt 499–4392)	[31]
	RUBrep/GFP_RdRp*	RUBrep/GFP replicons with changes in the putative RNA-dependent RNA polymerase catalytic domain at Asp1967 to Lys and Asp1968 to Leu	PCR amplification with mutagenic primers and swapped the Bgl II-Fse I fragment (nt 5355–6091)	
RuV mini-Xpress system	g41-GFP (or g41-Rfp)	RuV genomic mini replicon with RuV 5′ 41-nt fused with the GFP (or RFP) gene followed by 3′ terminal 400 nts (or 600 nts for g41-Rfp)	Replacing the EcoN I-EcoRI fragment from Robo402 with the PCR amplified subgenomic region of RUBrep/GFP (or RUBrep/Rfp) (with EcoN I-EcoR I sites)	
	g1700-GFP (or g1700-Rfp)	RuV expression system RuV 5′1692-nt fused with the GFP (or RFP) gene followed by 3′ terminal 400 nts (or 600 nts for g1700-Rfp)	Replacing the Not I-EcoRI fragment from Robo402 with the PCR amplified subgenomic region of RUBrep/GFP (or RUBrep/Rfp) (with Not I-EcoR I sites)	
	pUC-sg-GFP (or pUC-sg-Rfp)	RuV expression system containing the subgenomic sequences of RUBrep/GFP (or RUBrep/Rfp)	PCR amplification of subgenomic RNA sequences from RUBrep/GFP (or RUBrep/Rfp) and clone to pUC18 vector; the forward primer contains Hind III site and SP6 RNA polymerase promoter	
Controls	pCI-GFP (or RFP)	GFP (or RFP) control plasmid in pCI-Neo vector (Promega)	PCR amplified GFP (or RFP) gene was cloned to pCI-Neo vector (Promega) between Nhe I-EcoR I sites	
	pGEM-GFP	PCR controls or probe syntheses; in pGEM3Zf(−) vector (Promega)	PCR amplified GFP gene was cloned to pGEM3Zf(−) vector (Promega) between HindIII-EcoR I sites	

## Data Availability

The data presented in this study are available on request from the corresponding author.

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
