# Peer review of "Exogenous Rubella Virus Capsid Proteins Enhance Virus Genome Replication"

_pathogens, 2022, doi:10.3390/pathogens11060683_

Round 1

Reviewer 1 Report

In this study, the authors explored the role of rubella virus (RuV) capsid protein (CP) in virions on genome replication enhancement during the early stage of the virus infection cycle. The authors found that exogenous RuV CP enhanced virus genome replication. The authors also showed that CP did not affect the translation efficiency from genomic or subgenomic RNA, the intracellular distribution of the non-structural proteins, or subgenomic RNA synthesis. They suggest that the exogenous RuV CP increases the efficiency of early viral genome replication by modulating the stage(s) prior to and/or at the initiation of negative-strand RNA synthesis.

The findings of this study are relevant to the field of RuV, especially in contributing to understanding the RuV infection life cycle.

In general, the manuscript has been well written. However, I have some minor comments as the following

1/. In my opinion, the main question should appear in the abstract. Therefore, one sentence about the study question, the reason for the study, etc., should be added in line 13, page 1.

2/. To be easier for the general reader, an abbreviation of the GFP should be added, probably on page 2, line 52

3/. Please correct the doi address of reference No. 10, page 18, line 653.

Reviewer 2 Report

The authors have previously shown that intracellularly supplied rubella viral capsid protein (CP) promotes the viral genome replication. Based on the previous finding, they describe that it can be exerted by CP supplied exogenously in this manuscript. They also show that in the presence of CP, translation efficiency from the viral genome, subcellular distribution of nonstructural proteins, and sgRNA synthesis are not affected, while the rubella virus genome is specifically retained. This suggests that increased sRNA stability affects the initial process of RuV genome replication.

This new finding suggests that RuV CP released from RuV particles in infected cells may directly contribute to the genome replication. This function of CP is unique to rubella virus among many plus-strand RNA viruses and is potentially of great interest. However, there are many critical problems in this paper, as follows, and it needs to be extensively modified to prove the authors' claims.

Major comments

  1. In many experiments, RuV antigens and recombinant proteins were added during RNA transfection, but where these antigens work (intracellularly or extracellularly?) has not been examined and remains unclear. Additional experiments or detailed discussions are needed to clarify this issue.
    Viral particles (UV-inactivated RuV and VLP) and recombinant protein (rCP) may have fundamentally different places of function: in the case of viral particles, they are expected to enter the cell using the normal RuV entry pathway (receptor-dependent pathway), whereas in the case of recombinant proteins (especially CP), the pathway is not expected to be available. How are the recombinant proteins internalized into cells if they function intracellular? The transfection reagent used in this study is generally not used to introduce proteins to cells.
    If extracellular, how do they work? The authors suggest that CP may improve RuV RNA retention by protecting it. Does CP directly bind to the RNA before it is taken up into the cells? In that case, Why can CP added 4 hours after transfection still promote genome replication, as shown on page 5, line 178?

  2. Figure 4B and 4C: All images are cropped and discontinuous. They should be retaken as single continuous images or the cut lines should be clearly shown. Compared to the original image attached, the image for RT-PCR to detect the negative strand RNA seems to use different images between +rC and +rE2. (The +rE2 image was not provided). In this case, the comparison between lanes rC and +rE2 is particularly important and must be displayed in one continuous image. In the image for RT-PCR to detect the positive strand, only lane for the molecular weight marker appears to have been added over from a different image. These kinds of image processing make one suspect research misconduct, and I am deeply disappointed that I have to point this out.

  3. Figure 6: Replication-competent replicons are not appropriate for the purpose of this analysis for RNA stability, because they are themselves replicated. Only non-replicative replicons should be evaluated and the results of the WT, NSP-RFP and 1301S should be removed from this figure to avoid misunderstanding. The sentence "although a more dramatic difference occurred among replication-competent replicons” in the Abstract also should be removed.
  4. The relative amounts (rCP/rE2) of NSP-ATG* and RdRp* RNAs are shown to be 1.2±0.2 and 1.1±0.2, respectively. This result seems to indicate that there is no difference in the effect of E2 and CP on RNA stability. The authors show that there is a significant difference compared to the control RNA ratio (rCP/rE2), but this does not prove that rubella virus RNA specifically increases stability. For example, CP or E2 may just affect the stability of the control RNA. Any explanations? The result of the experiment described in Pages 10-11, Lines 362-367, may help to support the authors' claim regarding to RNA stability. Please display this data (in a 4d timeframe) in a figure.
  5. Figure S1: The authors conclude from the results in Fig. 6 that CP improves retention of RuV RNA. Thus, it might be expected that translation from genomic RNA/subgenomic RNA would also be improved. However, no such improvement in translation has been observed. Please discuss this.

Minor points

  1. Most of the rubella virus antigens used in the experiment of addition (UV-inactivated RuV antigen, VLP, and recombinant proteins) were purchased from manufactures. Since these are not common reagents, descriptions are required in detail. Although the UV-inactivated RuV has been subjected to freeze-thaw treatment by the authors, what are purpose and result of this (RuV is relatively stable for freeze-thaw cycles). Please define VLP. For recombinant proteins, it should be stated about at least amino acid lengths (are they full-length?) and which expression system they are prepared.

  2. Figure 4B: To ensure that the gRNA and sgRNA bands are properly shown in the Northern blotting images, a molecular weight marker or reference RNA should be simultaneously migrated with samples.
  3. Table 2: GFP expression should be determined quantitatively, not by Yes or No. The table should also show GFP expression without rCP.

  4. Table 3: In the field of virology, "replicon" is generally thought to refer to self-replicating viral RNA. The authors refer to RNAs with viral 5' and 3' end sequences as "mini-replicons," but to avoid confusion, it is better to use a different name.

  5. Supplementary Figure 2: The explanations for panels B and C are reversed. Also, the explanations for the right and left of panel B are also reversed.

  6. 4.3 Constructs and Transfection: Sabin poliovirus 2 is currently strictly controlled for use under GAPIII. Please state that it is used in accordance with the regulations.

  7. Line 62: Please check for a reference format of the paper by Prasad et al.

  8. Line 105: “Advanced ImmunoChemicals” may be a typo of “Advanced ImmunoChemical”.
  9. Line 309: As far as I know, cytomegalovirus does not have an IRES sequence. It may be mistaken for Encephalomyocarditis virus (EMCV).

Reviewer 3 Report

Review

Exogenous Rubella Virus Capsid Proteins Enhance Virus Ge-nome Replication by Chen et al.

The authors present a study that investigates the function of the capsid protein of rubella virus. C protein contains several arginine and proline residues resulting in a net positively charged protein. C protein assembles therefore with the negatively charged viral RNA to form the viral nucleocapsids. Moreover, C is a multifunctional protein that interacts with the cytoplasmic tails of E1 and E2 and with two mitochondrial proteins, p32 and Par-4. C performs several functions in viral assembly, transcription and replication.

Rubella virus causes a mild disease in unvaccinated children and adults, while infection in unprotected pregnant women can result in dramatic malformation and death of the unborn child, known as congenital rubella syndrome, CRS. Despite the fact that a safe and efficient vaccine exists, 100,000 children are estimated globally to be affected by CRS. Rubella virus is definitely a public health issue and should be studied e.g. in comparison with other teratogens for improved understanding of the molecular causes of damage during embryonic development.

The authors found that exogenous RuV CP enhanced viral replication and accumulated negative-strand RNA genomes in transfections supplemented with exogenous recombinant CP (rCP). They demonstrated that CP does not influence the efficiency of translation from the genomic or subgenomic RNA. Likewise, the intracellular distribution of the non-structural proteins, or synthesis of the sgRNA was not affected. When mutants of the RuV genome were investigated to map the domains responsible for this effect and thereby hopefully gathering information with respect to its nature, rCP was found to restore replication in constructs bearing a mutation in the NSP ORF but failed to fully support replicons with a defect in the synthesis of the positive-strand RNA. The authors conclude that the exogenous RuV CP increases efficiency of early viral genome replication by modulating stages prior to and/or at the initiation of negative-strand RNA synthesis, possibly through a more generally acting mechanism such as protection of viral RNA molecules.

This is a clear and sound study that uses well-established methods in molecular virology. It adds to the understanding of the functions of C-protein.

General comments:

It was very difficult for me to follow all the different constructs. A figure in combination with tables 2 and 3 showing the constructs and their features would be helpful.

I have difficulties to understand the term retention and therefore the meaning and results of the respective experiments. The M+M section was also not helpful to understand that issue. The authors should consider to further elaborate on that point.

Specific comments:

Page 1, line 34: I suggest to change the order to “E1 and E2”

Line 40: omit putative, not necessary in this context

Page 2, line 90 ff: I have not found information with respect to the pseudovirus und the meaning of P1 and P2 in lines 95 and 96

Page 3, line 112: the two new Matonaviruses RuhV und RusV would be interesting in addition to Sindbis virus :-)

Line 124: I suggest to list the names of the antigens explicitly

Page 6, line 231: There is no panel with cells at 48-h post transfection in Fig. 3A. I found only 18 h and 5 d...

Page 11, line 385 and Page 12, line 419: please check the wording of the sentences

Round 2

Reviewer 2 Report

All comments have been responded to honestly, and there has been improvement in the content. However, there are still a few points that need to be confirmed.

(1) Fig. 6A:

The revised manuscript (Lines 446-449) states “The total fluorescence from each timepoint was normalized to the fluorescence at day 0. We found that, at 24-h post-transfection, there were 1.6-fold more signals from RuV RNA species if the RNA was mixed with rCP rather than with rE2, while this ratio was ~0.9 with control RNA (Figure 6A)”. However, according to the graph, it reads about 0.8 for rCP and about 0.6 for rE2, from which the ratio is calculated to be about 1.3-fold. Also, in the case of the control Rfp RNA, the value in the description and that in the graph do not seem to match. Please confirm if the description in the text is correct.

(2) Lines 499-500:

In the experiment in Fig. 6b, the replication-competent replicon is RUBrep/GFP (WT) only. However, a range of values for several replicons is shown in the text. Please correct this. Please also provide a statistical analysis of the results for the control RNA and rubella virus RNA values at 6 hours post transfection.

(3) Supplementary figure 3

Correct the description of the right half of the NSP in the genome structure diagram, from p150 to p90.
